# Review of Photodetectors for Space Lidars

**DOI:** 10.3390/s24206620

**Published:** 2024-10-14

**Authors:** Xiaoli Sun

**Affiliations:** Planetary Geology, Geophysics and Geochemistry Laboratory, Solar System Exploration Division, NASA Goddard Space Flight Center, Greenbelt, MD 20771, USA; xiaoli.sun-1@nasa.gov

**Keywords:** avalanche photodiode, photodetectors, lidar, space, remote sensing

## Abstract

Photodetectors play a critical role in space lidars designed for scientific investigations from orbit around planetary bodies. The detectors must be highly sensitive due to the long range of measurements and tight constraints on the size, weight, and power of the instrument. The detectors must also be space radiation tolerant over multi-year mission lifetimes with no significant performance degradation. Early space lidars used diode-pumped Nd:YAG lasers with a single beam for range and atmospheric backscattering measurements at 1064 nm or its frequency harmonics. The photodetectors used were single-element photomultiplier tubes and infrared performance-enhanced silicon avalanche photodiodes. Space lidars have advanced to multiple beams for surface topographic mapping and active infrared spectroscopic measurements of atmospheric species and surface composition, which demand increased performance and new capabilities for lidar detectors. Higher sensitivity detectors are required so that multi-beam and multi-wavelength measurements can be performed without increasing the laser and instrument power. Pixelated photodetectors are needed so that a single detector assembly can be used for simultaneous multi-channel measurements. Photon-counting photodetectors are needed for active spectroscopy measurements from short-wave infrared to mid-wave infrared. HgCdTe avalanche photodiode arrays have emerged recently as a promising technology to fill these needs. This paper gives a review of the photodetectors used in past and present lidars and the development and outlook of HgCdTe APD arrays for future space lidars.

## 1. Introduction

Light detection and ranging (lidar) is an important tool for remote sensing of planetary bodies from orbits. Starting with the Clementine LIDAR in 1994, there have been a series of space lidars for scientific investigations over the past 30 years, and several new ones currently in space [1,2]. Photodetectors play a vital role in space lidars where the spacecraft altitude is high, and instrument mass and power must be kept as low as possible. Unlike imaging sensors, lidar detectors must provide instantaneous signal outputs in response to the target returns instead of reading them out frame by frame. Lidar detectors need to have high photon-to-photoelectron conversion efficiency, also known as quantum efficiency (QE), at the laser wavelengths used by the lidar. Single-photon detectors are desirable to attain the highest receiver sensitivity. A linear response is needed for atmospheric backscatter profile and surface reflectance measurements. A wide dynamic range is required to accommodate all received signals from atmospheric backscatter, ground returns through clouds, and ground returns from different types of surfaces. For topographic measurement, multi-pixel photodiode arrays are needed in order to map the surface with a contiguous swath.

There have been only a few types of photodetectors used in space lidars to date. Photomultiplier tubes (PMTs) have been used for ultraviolet to visible laser wavelengths. Silicon avalanche photodiodes (APDs) in linear mode have been used for visible to near-infrared wavelengths. InGaAs APDs in linear mode have been used for near-infrared to short-wave infrared (SWIR) wavelengths. The performances of these detectors were limited by their specific technologies. The avalanche gain of an APD is a random variable, which results in excess noise in the output signal. The excess noise increases with the APD gain, which becomes overwhelming at a certain point. The useful gain of the silicon and InGaAs APDs in linear mode operation is still too low for the photocurrent to completely override the electronic noise of the preamplifiers. They cannot detect individual single-photon events but a few hundred photons in a short laser pulse. APDs in Geiger mode, also known as single-photon avalanche photodiodes (SPADs), have sufficiently high gain to detect single photons, but they cannot resolve the number of photons in a detected pulse and have a dead time after each detection. SPADs also have afterpulsing after an avalanche breakdown. To minimize this effect, the APD must be gated off long enough for the trapped electrons to die out. It is difficult to use gated photodetectors in lidars since they require the lidar receiver to predict the target range before the measurement. Gated photodetectors are also difficult to use for atmospheric backscatter measurements since the received signal is distributed over a long time period as the laser pulses propagate through the atmosphere.

Multi-pixel HgCdTe APD arrays have recently emerged as the detector of choice for lidars in SWIR and mid-wave infrared (MWIR) wavelengths [3]. They give linear analog waveform output over a wide dynamic range. The gain of HgCdTe APDs is nearly deterministic, with little excess noise added to the output photocurrent. The useful APD gain is sufficiently high for the photocurrent to override the electronic noise. The detectors are capable of achieving near quantum-limited performance where the dominant noise is shot noise from the quantum nature of photon detection. HgCdTe APDs have shown outstanding performance in several airborne lidar demonstrations.

This paper is intended as a review of the photodetectors used in past and current space lidars from visible to SWIR wavelengths, as well as a discussion of the latest advancement in HgCdTe APD arrays which fill the gap of single photon detectors from SWIR to MWIR wavelengths. This review is broken down by wavelength range and detector technology. The discussion of each detector technology includes its use in previous or current space missions, a brief review of the physics of photon detection and amplification, details on the electrical characteristics and modes of operation, and the effects of space radiation.

## 2. Space Lidar Detectors from UV to near Infrared

Most of the space lidars to date used diode-pumped Q-switched Nd:YAG laser at the 1064 nm fundamental lasing wavelength and/or the 532, 355, or 266 nm frequency harmonics. These lasers give high-output pulse energy and can operate in space for multiple years without being serviced. The photodetectors used in these space lidars were PMTs for the 266, 355, and 532 nm wavelengths [4] and silicon APDs for the 532 and 1064 nm wavelengths [2].

### 2.1. Photoelectron Multiplication Tubes

Ruggedized PMTs have long been used in space for passive spectrometers and lidars. They are used in the Lidar In-space Technology Experiment (LITE) on the Space Shuttle launched in 1994 [4], the Cloud-Aerosol Lidar with Orthogonal Polarization (CALIOP) on the Cloud-Aerosol Lidar and Infrared Pathfinder Satellite Observation (CALIPSO) mission launched in 2006 [5], the Atmospheric Dynamics Mission Aeolus (ADM-Aeolus) Doppler wind lidar in 2018 [6], and the Advanced Topographic Laser System (ATLAS) on the ICESat-2 mission launched in 2018 [7,8].

A PMT consists of a photocathode followed by a series of dynodes in a glass or ceramic housing [9]. The first PMTs were demonstrated almost 100 years ago. They have since been widely used in passive spectrometers on multi-year space missions. The internal photoelectron multiplication gain is in the range of from 10^5^ to 10^7^. The QE depends on the photocathode materials [9]. Bi-alkali photocathodes have 10–30% QE over the 160–600 nm wavelength range. Multi-alkali photocathodes have a lower QE but are sensitive up to 900 nm. GaAsP photocathodes have a near 40% QE from 450 to 650 nm. PMTs are mostly used in the visible spectral range. There are also InP/InGaAs photocathodes, which have a few percent QE up to 1600 nm but with much higher dark count rates. They require cooling to below −60 °C. There are also hybrid PMTs where the primary photoelectrons are focused and accelerated to impact an APD, which generates tens of thousands of secondary electrons at the impact and through the avalanche process [10]. The InP/InGaAs photocathode and hybrid PMTs are relatively new devices and need further development for possible space applications. Segmented anode PMTs are now available, which can be used as multi-pixel array sensors. Microchannel plates (MCPs) can be used in place of the dynodes to provide high gain in a much smaller package and with pixelated output when used with segmented anodes.

PMTs are usually used in single-photon counting mode. The photoelectron multiplication gain is sufficiently high to completely override electronics noise from ordinary radio frequency (RF) amplifiers. A discriminator is used to distinguish single photon events. The dark count rates are typically less than 100 counts per second. The discriminator needs a recovery time after each triggering, which causes a dead time in the photon detection and limits the maximum photon count rate to a few hundred million per second. The PMT dead time does not seriously affect the lidar performance when the total signal and noise photon rate is low. However, the dead time imposes a major limitation on surface elevation and vegetation lidars with multiple photons in the received pulses. One method to mitigate this nonlinear effect is to use segment anodes and spread out the incident optical signal over multiple pixels, as in ICESat-2/ATLAS [11]. Figure 1 shows photographs of the PMTs used in ATLAS.

PMTs can also be used in analog mode by digitizing and recording the output pulse waveforms rather than counting the pulses, as in CALIPSO [5]. The primary photoelectrons are continuously multiplied as they arrive. There is no dead time since there is no breakdown in the photocathode and dynode operations. There is, however, additional pulse amplitude noise due to the randomness of the PMT gain. The average PMT gain can be adjusted by changing the bias voltage. However, the range of adjustment is limited since lowering the cathode and dynode voltage can result in a loss of the photoelectron collection efficiency by the anode. The randomness of the PMT gain is usually characterized by the excess noise factor, which is defined as the ratio of the mean squared to the square of the mean of the PMT gain. The excess noise factor of PMTs in analog mode is about 1.25 [12].

A major advantage of PMTs is the combination of large detector active area and wide electrical bandwidth, which is important for space lidars, which need fast time response and a relatively wide field of view (FOV). PMTs also have afterpulsing, which is caused by the ionization of residual gas molecules inside the PMT after a relatively strong optical input. Positive ions can hit the photocathode and produce spurious output after the incident optical signal. However, the afterpulsing amplitude and rate of occurrences in modern PMTs are relatively low. They do not usually cause problems in the lidar measurement.

PMTs are far less susceptible to space radiation damage compared to semiconductor devices. Major radiation effects on PMTs are scintillation and darkening of the glass window and housing [13]. Scintillation is a transient effect that produces spurious pulses at the PMT output. Window darkening is a permanent effect that causes a reduction in the incident optical signal power. Another radiation effect is the transient response to high-energy particles that hit the photocathodes and dynodes. There is no permanent damage from these transient events except for spurious noise outputs as the spacecraft flies through high-radiation regions, such as the South Atlantic Anomaly (SAA) [5].

PMTs are known to have a limited lifetime. The photocathodes can be damaged gradually by the ion bombardment from the residual gases during the photoelectron multiplication process. PMT lifetime mainly depends on the amount of electron charge drawn from the photocathodes. The lifetime of PMTs with MCP was generally shorter than regular PMTs in the past but has improved significantly in recent years [14]. The lifetime of modern PMTs is usually adequate for a multi-year space lidar mission, as has been demonstrated by ATLAS on the ICESat-2 mission.

### 2.2. Silicon Avalanche Photodiodes in Analog Mode Operation

Silicon APDs have been used in various lidars from visible to near-infrared wavelengths. They can be operated in either linear mode to give analog pulse waveform output or Geiger mode to detect single photon events. In analog mode, the APD bias is kept well below the breakdown voltage so that the devices provide a sustained multiplication gain for the primary photoelectrons.

The infrared QE-enhanced silicon APDs from Excelitas Canada Inc. (formally RCA, GE, EG&G, and PerkinElmer Canada), Montreal, QC, Canada [15,16] have been the detector of choice at 1064 nm laser wavelength for almost all space lidars to date. Since silicon has a low photon absorption coefficient at near-infrared, these APDs have a relatively thick active volume (0.12 mm) to increase the photon absorption path length. The active area of the APDs is 0.7 mm in diameter and contains an array of small dimples to deflect the incident light to further increase the light path length. These silicon APDs have a near 40% QE at 1064 nm at room temperature and can detect 100–200 photon laser pulses. The QE at 1064 nm increases with the temperature [15]. Since the APD dark current also increases with temperature, there is a trade-off between QE and dark current when choosing the APD operating temperature. The APD chip is integrated with the preamplifier on a hybrid circuit to minimize stray capacitance. The hybrid circuit is housed in a TO-8-type metal package and hermetically sealed with dry nitrogen. The preamplifier in the early devices was a high input impedance low-noise linear amplifier followed by a buffer amplifier. There was also a high voltage regulation and temperature compensation circuit on the same hybrid circuit. The APDs require from 300 to 500 V bias voltage to achieve the required gain (~100). The high voltage temperature compensation circuit automatically adjusts the bias to the APD to maintain a near-constant responsivity over the specified temperature range. Since the high voltage can cause corona at low pressure, the leak rate of the TO-8 package must be sufficiently low to maintain the nitrogen pressure over the mission lifetime.

These infrared performance-enhanced silicon APD modules were first developed for free space laser communications in the 1980’s. Several residual detector modules were used in the laser altimeter on the Clementine mission in 1994 [17], the Near-Earth Asteroid Rendezvous (NEAR) mission in 1996 [18], Mars Oribter Laser Altimeter (MOLA) on the Mars Observer mission in 1992 [19], and MOLA-2 on the Mars Global Surveyor (MGS) in 1996 [2].

A new set of detector modules was made for the Geoscience Laser Altimeter System (GLAS) on the ICESat mission [2]. It contained the same silicon APD chips from the original lot, the same high-voltage regulator and temperature compensation circuit, and a transimpedance (TIA) preamplifier. The use of TIAs greatly improved the linearity, dynamic range, and pulse waveform fidelity. Figure 2 shows photographs and micrographs of the Si APD used in GLAS on the ICESat mission. These detector modules were also used in several subsequent space lidars, including the 1064 nm channel of the CALOIP lidar on the CALIPSO mission from 2006 to 2023 [5], the Mercury Laser Altimeter (MLA) on the MESSENGER mission from 2004 to 2014 [2], and the Lunar Orbiter Laser Altimeter (LOLA) on the LRO mission from 2009 to the present [2]. Another set of detector modules was made for the BepiColombo Laser Altimeter (BELA) on the BepiColombo mission to Mercury by the European Space Agency (ESA). BepiColombo was launched in October 2018 and is currently in its cruise phase to the planet [20]. A thermal electrical cooler (TEC) was used in the detector modules for BELA to maintain the APD operation temperature in a harsh thermal environment. The internal high-voltage regulator circuit was removed so that the APD gain could be adjusted to increase the dynamic range. A new set of detector modules was made for the Global Ecosystem Dynamics Investigation (GEDI) lidar on the International Space Station (ISS) from 2018 to the present [21]. The same APD chips and TIA are used in the GEDI detector modules, but a heater instead of a TEC is used to maintain the APD operation temperature. The APD chip is heated to nearly 70 °C to optimize the QE at the 1064 nm laser wavelength. A heater is more efficient and simpler to use than a TEC to set and maintain the APD chip temperature above the ambient temperature.

Commercial versions of the same Si APD were successfully used in the laser altimeters on the Hayabusa-1 and Hayabusa-2 missions to asteroids Itokawa and Ryugu [22], launched in 2003 and 2014, respectively. They were also used in the Ganymede Laser Altimeter (GALA) on the Jupiter Icy Moons Explorer (JUICE) mission launched in April 2023 [23]. The noise equivalent power (NEP) of commercial devices is typically two to three times higher than that of custom devices for space applications. Commercial devices also do not have a guard ring around the active area, which helps to reduce the dark current from the space radiation damage. Commercial devices use generic electrical, electronics, and electromechanical (EEE) parts, while space devices use all space-qualified EEE parts. A custom lot with additional part screening and testing is generally required when using commercial-grade devices for space instruments.

Si APDs are radiation damage tolerant [24]. The major radiation effect is the displacement damage by the impinging particles to the crystal and the creation of defect sites, which causes the dark current to increase. There are also transient effects as high-energy particles pass through the APD active volume. The APD produces a large output pulse equivalent to hundreds of thousands of photons in response to each transient particle through the device, which can briefly saturate the APD and interrupt the measurement. Lidar detectors are usually well shielded from space radiation by optics in front, a lens housing on the side, and electronics on the back. The total radiation dose and fluence at the detector chip are much lower than at other parts of the instrument. For example, the total dose at the detector chips of the GLAS silicon APDs was 3–5 krad(Si) over a five-year mission based on a 3-D ray-tracing analysis. We observed little receiver performance degradation due to radiation damage to the silicon APDs in MOLA, MLA, and GLAS during their entire missions [24]. There has been a steady increase in the detector dark noise from the five silicon APD modules on LOLA at 2 to 3% per year [24]. The LOLA detector assemblies are attached to the exterior of the instrument chassis. The amount of shielding is much thinner compared to the previous space lidars.

The analog silicon APD preamplifier modules from Excelitas are still the preferred detectors at 1064 nm laser wavelength and are likely to be used in space lidar for years to come. However, they are still far from single photon sensitivity. They are single-element detectors which are not power- and mass-efficient for use in multi-beam lidars or multi-pixel imaging lidars.

### 2.3. Silicon APDs in Geiger Mode Operation

Silicon APDs can also be biased above the breakdown voltage to operate in Geiger mode. An incident photon or a dark current electron can trigger an avalanche breakdown, resulting in a large electrical pulse at the output well above the electronic circuit noise [25,26,27]. The avalanche needs to be quenched after each photon detection by lowering the bias voltage to below the breakdown point. There are two types of quenching circuits: passive and active. The former relies on a current-limiting resistor to load down the bias voltage to stop the avalanche. The latter actively lowers the bias voltage upon the detection of an avalanche. The bias voltage must be held low for a short time to reduce the afterpulsing before being restored for the next photon detection. Active quenching has a much faster recovery time and, consequently, a higher maximum count rate.

The NASA Goddard Space Flight Center (GSFC) has worked with Excelitas Canada to space qualify their silicon APD single photon counting module (SPCM) [28]. These detectors have been used for the 532 nm atmosphere backscattering measurement of the GLAS instrument on ICESat. They have a single-photon counting efficiency of as high as 70% at a 532 nm laser wavelength, an active quenching circuit with a dead time of less than 50 ns, a dark count rate well below 1000 counts per second, and a patented high-voltage current-limiting circuit to prevent damage from transient space radiation events. Figure 3 shows the SPCM used in ICESat/GLAS.

The reliability of the commercial version of the SPCMs was also greatly improved after ICESat. Several commercial-grade SPCMs were used in NASA’s Clouds Aerosol Transport System (CATS) lidar on the ISS. All of them performed as expected during the entire mission time from September 2014 to March 2016 [29].

Another type of Geiger mode silicon APD used in space is the SPAD from the Czech Technical University in Prague, Czech Republic [30]. These detectors need to be gated to maintain a low after-pulsing rate. A unique feature of these SPADs is the ability to resolve the number of photons in the received pulse from the onset speed of the avalanche breakdown. Such information can be used to correct the range bias caused by the variable input pulse amplitude, provided the pulse shape is known. These silicon SPADs were used in a few early space lidars, including the lidar on the Mars 96 mission by Russia and the Mars Polar Lander on Mars Surveyor ’98 mission by NASA [31,32]. They were also used in laser time transfer experiments in space.

The single photon counting detectors from Micro Photon Devices (MPDs) have been considered for use in space applications [32]. These SPADs have a slightly lower photon counting efficiency and longer dead time compared to those of the SPCMs, but they have a much lower timing jitter and do not need to be gated. The active quenching circuits and the silicon APD are integrated into a silicon chip. The active area of these devices is smaller, 50 μm in diameter, compared to the 170 μm of the SPCMs. They can also have an array of SPADs on the same chip, though the spacing between the active elements is relatively wide.

A relatively new class of devices called Silicon Photo-Multipliers (SiPMs) became commercially available in recent years. They consist of an array of groups (subarrays) of passively quenched and parallelly connected SPADs [33]. Each SPAD can detect a photon and produce an electrical pulse at the output. The output pulse waveform is the sum of the outputs from all the SPADs in the subarray. When multiple SPADs in the same subarray are illuminated, the output pulse amplitude is roughly proportional to the total number of detected photons by the subarray. SiPMs have a large active area, single photon sensitivity, and the ability to resolve the photon number in the pulse. SiPMs require a much lower bias voltage than PMTs. They are relatively low-cost to fabricate. However, SiPMs are still Geiger mode APDs with the same nonlinear behaviors, such as afterpulsing. They have not been used in space lidars.

Geiger-mode silicon APDs are more susceptible to space radiation damage since they are sensitive to single electron events [34]. Fortunately, the damage rate for silicon SPADs is still acceptable for a typical multi-year Earth science mission. For example, the dark count rate of the SPCMs on the ICESat mission increased from several hundred per second to several hundred thousand per second at the end of its 9-year mission [35], which was still much lower than the rate of solar background photons for daytime measurement.

## 3. Space Lidar Detectors from Short-Wave to Mid-Wave Infrared

There is an increasing demand for spectroscopic lidars at SWIR (1–2.5 μm) and MWIR (from 2.5 to 5 μm) wavelengths. There are many spectral absorption lines of interest in this wavelength region for both atmospheric chemistry and surface geology and geochemistry. The solar background illumination is much lower. The laser eye-safety threshold is much higher above 1.4 μm wavelength. InGaAs APDs are currently the detector of choice for SWIR lidars. There has been no MWIR lidar in space to date due to the lack of high-speed and high-sensitivity detectors. HgCdTe APDs have recently emerged as new lidar detectors for wavelengths from SWIR to MWIR. They have much better performance than InGaAs APDs in SWIR. They fill the gap of single-photon detectors in MWIR. NASA GSFC has been working with industry over the past several years to develop the HgCdTe APD technology for future space lidars.

### 3.1. InGaAs APDs

InGaAs APDs in linear mode have been used in space lidars for laser wavelengths up to 2 μm. They are used in the Methane Remote sensing Lidar Mission (MERLIN) at the 1.645 μm methane absorption line wavelength, which is to be launched in 2028 [36]. They were also used in the Lunar Flashlight, which is a multiwavelength reflectometer at 1.064, 1.495, 1.850, and 1.990 μm laser wavelengths for lunar volatile measurements in permanent shadow regions from low orbital altitudes of from 10 to 20 km [37,38]. Lunar flashlight was launched in December 2022. Although the spacecraft did not enter lunar orbit due to a malfunction of the propulsion system, the instrument with the InGaAs APD was functional after launch.

InGaAs APDs have a high QE (>60%) up to a 2 μm wavelength range. They are also relatively low-cost. However, the useful gain of InGaAs APDs is typically from 20 to 30 before the dark noise and APD gain excess noise become overwhelming. As a result, the dominant noise source is still the preamplifier. The combined NEP from the APD and preamplifier is much higher than that of the APD given in the datasheet. The dark currents of InGaAs APDs are also much higher compared to silicon APDs. Cooling the APD only reduces the dark currents but not the preamplifier noise. The NEP of the best InGaAs APD lidar receiver is in the range of 100–200 fW/Hz^1/2^, which is about 10 times that of linear mode silicon APDs with the same preamplifier. InGaAs APDs are also more susceptible to space radiation damage than silicon APDs [39,40].

InGaAs APDs in Geiger mode can detect single photons, but they have not been used in space lidars at present. The afterpulsing rate of InGaAs APD is much higher than that of silicon APDs. They must be gated in order to maintain an acceptable afterpulsing rate [41], which makes them more difficult to use in space lidars.

### 3.2. HgCdTe APDs for Atmospheric Lidars

HgCdTe is a semiconductor alloy with many properties suited for infrared photodetectors. It has a >90% quantum efficiency from SWIR to long-wave infrared (LWIR). The cutoff wavelength can be tailored by the HgCdTe composition. The dark current is lower than other types of photodiodes sensitive in this wavelength range. HgCdTe photodiode arrays have been used as imaging sensors for astronomy and infrared spectrometers for decades [42]. HgCdTe APD arrays became available in recent years [43,44]. The APD gain can be as high as several thousand, which is sufficient for the photocurrent to completely override the electronic noise of the preamplifiers. The APD gain is almost deterministic because of the ballistic nature of the electron multiplication process and the much lower efficiency for hole multiplication. They have demonstrated near-quantum limited performance in analog operation [45]. The outputs of HgCdTe APDs are like those from silicon APDs but with much lower noise. The major disadvantage of HgCdTe APDs is the low operation temperature. The APD has to be housed in a cryo-cooler. Early HgCdTe APD arrays had a relatively slow time response. The read-out integrated circuits (ROICs) are separate from the APD array. The overall electrical bandwidth is too low for high-precision ranging but adequate for atmospheric backscatter and spectroscopic surface reflectance measurements.

A set of 4 × 4 pixel HgCdTe APD arrays were designed and fabricated by Leonardo DRS for the integrated path differential absorption (IPDA) greenhouse gas lidars supported by the NASA Earth Science Technology Office (ESTO) [45]. The HgCdTe APD arrays are based on the high-density vertically-integrated photodiode (HDVIP) technology, which has a cylindrical structure for each diode, as shown in Figure 4. Each pixel consists of four APDs connected in parallel. The pixel size is 80 × 80 μm at 80 μm pitch. A guard band is placed around the pixel array, which is made of identical pixels electrically connected in parallel. The outputs from all the 4 × 4 pixels are fanned out to the side of the detector chip and wire-bonded to 16-channel TIAs in the ROIC. The ROIC chip is placed next to the APD array on the same ceramic carrier.

The spectral response of these HgCdTe APD spans from visible to 4.4 μm. Figure 5 shows a plot of the measured QE vs. wavelength [46]. The QE here is defined as the ratio of the number of output photoelectrons to the number of incident photons at unity APD gain. At short wavelengths, when the photon energy is many times the HgCdTe bandgap, there can be more than one pair of electron–hole generated from a single photon [47]. This is why the QE shown in Figure 5 is slightly higher than 100% near a 1 μm wavelength. Although this is an artifact resulting from our QE definition, the additional photocurrent from the extra hole–electron pairs are real. This definition of QE is widely used in the lidar equation to calculate detected signals.

The ROIC for these HgCdTe APD arrays is made of silicon and fabricated using a standard silicon integrated circuit foundry service. The ROICs are custom-designed such that they can be tested at room temperature but used with the APD array at a cryogenic temperature. The side-by-side arrangement allows the APD arrays and ROIC chips to be fabricated and tested separately, which reduces the risks and cost of the device production. The electrical bandwidth of the TIAs is about 8 MHz with a 20 ns pulse rise time and 50 ns fall time. The ROIC can be configured as capacitive transimpedance amplifiers (CTIAs), which integrate the photocurrent onto a small capacitor and reset periodically. The CTIA mode can be used to measure the APD output currents to below femtoamperes.

For greenhouse gas lidars, the laser beam divergence and the receiver FOV need to be relatively large to reduce laser speckle noise. The received optical signal illuminates several pixels of the HgCdTe APD array, and the outputs from selected pixels are summed together. The pixel selection depends on the focal spot size and location of the incident optical signal on the pixel array. The total dark noise is the root-sum-square of the noise from the pixels being summed. There is no dead space between pixels but four voids inside each pixel around the via of each diode. The overall fill factor of the photon detection area is 75% at the maximum APD gain.

Table 1 lists the major parameters of these 4 × 4 pixel HgCdTe APD arrays at 80 K. The most noticeable difference in the device parameters compared to other types of APDs is the NEP, which gives a measure of the minimum detectable optical signal power. The NEP for a typical InGaAs APD with a low-noise preamplifier in a hybrid module (e.g., Excelitas LLAM-1550-R08BH) is about 100 fW/Hz^1/2^, whereas the NEP for these HgCdTe APDs is <0.5 fW/Hz^1/2^.

A unique property of HgCdTe APDs is that the gain is an exponential function of the bias voltage. In the case of HDVIP APDs, the gain increases by roughly a factor of two for every volt increase in the bias above 6 V, which makes it easy to adjust and control. The APD gain is equal to unity at <1.0 V bias, which makes it easy to measure the quantum efficiency. The gain-normalized dark current remains the same over the entire gain range until the trap-assistant tunneling starts to increase rapidly. The devices have a very wide linear dynamic range. The low end of the dynamic range is limited by the detector noise floor, which is several hundred μV per pixel in standard deviation. The highest output signal level is 1.1 V, which is limited by the TIA. The detector output is usually averaged over hundreds of successive measurements in an atmospheric lidar, which further lowers the noise floor and increases the dynamic range. In addition, the APD gain can be adjusted by a factor of nearly 1000, and the TIA gain can be adjusted by a factor of 7 for these devices, which further extends the receiver dynamic range.

The major complication of using HgCdTe APDs is the need for cryo-coolers. The APD array must be held below 135 K during operation, which is beyond what a multi-stage TEC can achieve. There are several small cryo-cooler designs available for small detectors in space instruments. One example is the Micro-cooler, which is a miniaturized version of the high reliability and long lifetime pulse-tube coolers [48]. A low-cost alternative is tactical coolers, which are ruggedized mini-Stirling coolers used in night-vision equipment. These tactical coolers have demonstrated lifetimes of tens of thousands of hours [49]. Leonardo DRS has successfully built and tested an integrated detector cooler assembly (IDCA) with a 4 × 4 pixel HgCdTe APD array in a tactical cooler, as shown in Figure 6. The IDCA uses < 8 W electrical power, including peripheral electronics, and weighs 1.2 kg, including the support structure. The detector can be cooled to 80 K within 7 min.

### 3.3. HgCdTe APD Arrays for Surface Elevation Lidar

HgCdTe APD arrays with higher electrical bandwidth and even lower noise have become available [50]. They can detect single photon events and be used for linear mode photon counting (LMPC). The photocurrent pulse from a single photon detection can rise above the noise floor and be readily distinguished. These devices are available now in 2 × 8 pixel arrays from Leonardo DRS. The detector output is the linear sum of the pulse waveforms from individual photon detections. A waveform digitizer can be used for data acquisition. There is no need for a discriminator and no dead time after each photon detection. There is also no afterpulsing.

The ROICs for the LMPC HgCdTe APD arrays are embedded in the silicon substrate of the APD or connected to the APD array via indium bumps to minimize stray capacitance. The noise from the TIAs is much lower than the noise from RF preamplifiers used for PMTs. Therefore, the APD gain required to detect single photons is several orders of magnitude lower than that of the PMTs.

The first LMPC HgCdTe APD array reported was a 2 × 8 pixel HDVIP HgCdTe APD array with a cutoff wavelength of 4.4 μm [50]. The device structure is the same as that shown in Figure 4, except that the ROIC is embedded in the silicon substrate. The pixel size was 64 × 64 μm at 64 μm pitch. The optimal region for single photon detection was the center area in between the four diodes in a pixel because the photoelectrons could transverse through the entire APD multiplication regions. The diameter of the optimal detection area was about 22 μm. The APD gain was about 500 at 12 V bias in the earlier devices, which was sufficient for single photon detection. The output pulse width in response to a single photon detection was 6–9 ns. The average pulse amplitude was at least 10 times the noise standard deviation from the TIA. The maximum photon counting efficiency (PDE) was measured to be 60% at a false event rate (FER) of about 10^6^ counts/s per pixel. The FER was found mostly from the photon emission from the ROIC, which was largely suppressed in later devices.

A new batch of 2 × 8 pixel devices was fabricated for NASA GSFC under an ESTO Advanced Component Technology (ACT) program from 2012 to 2015 [51]. The dark count rate was reduced by about a factor of 10 by adding a metal layer between the ROIC and the HgCdTe APD array to block photons emitted from the ROIC. The APD gains were as high as 1900 in some of the devices. The reliability and yield of the devices were also greatly improved. Two IDCAs were built under a NASA ESTO In-space Validation of Earth Science Technology (InVEST) program for potential use on a CubeSat. Both the APD arrays and the IDCA were fully tested and characterized at both Leonardo DRS and NASA GSFC [51]. Two new 2 × 8 pixel HgCdTe APD IDCAs were delivered to NASA GSFC recently, which had a much improved cold-shield design and lower dark noise. Figure 7 shows the latest measurement results of count rates vs. detection threshold with the detector in dark and under illumination at a 1.03 μm laser wavelength. The PDE is calculated as the ratio of the net detected photon count rate to the incident photon rate at the given detection threshold. The FER is the count rate with the detector in the dark. The same tests were performed at a 1.55 μm wavelength. The measurement results were about the same as those at 1.03 μm.

The QE of the device was also measured directly from the analog signal amplitude in response to known peak power incident laser pulses. The measured QEs from the analog pulse waveform and the maximum PDEs from photon counts at 1.55 μm wavelength for all 16 pixels are plotted in Figure 8. The QEs from the analog signal are slightly higher than the PDEs since the pulse amplitudes from some of the detected photons were lower than the detection threshold.

The detector impulse response was measured with a <0.1 ns pulse width laser. The output pulse width was 9 ns at full-width half maximum (FWHM) at the maximum APD bias (12 V) and the maximum TIA gain setting. The pulse width becomes 6–7 ns FWHM at the minimum TIA gain setting. Lower TIA gain improves the electrical bandwidth but slightly reduces the signal-to-noise ratio (SNR). The output pulse shape and pulse amplitude vs. input pulse energy at maximum TIA gain are plotted in Figure 9. It shows that the detector has a useful dynamic range of 60 photos/pulse and 100 photons/pulse for two TIA gain settings. A wider dynamic range can be obtained by lowering the APD gain at the cost of a higher noise contribution from electronics.

The optimal detector active area of these LMPC HgCdTe APD arrays is smaller than the size of the pixel. Microlens arrays can be used to concentrate the signal onto the center of each pixel to improve the fill factor. The microlens array is placed one focal distance above the detector, and the optical signal is focused on the top of a microlens array [52]. Microlens arrays were implemented in all 2 × 8 pixel IDCAs built for NASA GSFC. Figure 10 shows a photograph of the microlens array on top of the HgCdTe APD array and the surface scans of the net photon count rate vs. the position of the laser spot. The microlens array on the 2 × 8 pixel APD array has a focal length of 60 μm. The numerical aperture of a pixel with the microlens array reduces by about f/3, which is more than sufficient for a high-resolution surface elevation swath mapping lidar.

The 2 × 8 pixel IDCAs have gone through full function, performance, and environmental tests. It was also found that the APD arrays can be operated at 100 K with the same performance as at 80 K. Figure 11 shows photographs of two 2 × 8 pixel IDCAs with different arrangements of the compressor and the cooler. Table 2 lists the characteristics of the latest IDCA.

The development of HgCdTe APD arrays continues to evolve. Leonardo DRS continues to improve its HDVIP device performance, reliability, and yield. A new batch of devices with a 3.7 μm cutoff wavelength has been fabricated and showed similar performance. A maximum APD gain of 6000 has been measured from the 4 × 4 pixel devices without a significant increase in dark current. A 2 × 30 and a 7 × 8 pixel LMPC HgCdTe APD array have been built and tested. The bandwidth of the TIA has been increased by further reducing the stray and parasitic capacitance of the ROIC. The impulse response pulse width has been reduced to about 3.5 ns FWHM. More details about their progress can be found in Anderson et al. (2022) [46].

Leonardo UK has been developing HgCdTe APD arrays for imaging sensors for astronomy. They are also developing HgCdTe APD arrays for linear mode photon detection, active imaging, and lidar applications. These APD arrays have a planer structure with a 100% fill factor [53,54,55]. The cutoff wavelength of the photon absorption layer is 2.5 μm. The highest reported APD gain is 446 at a 17.7 V bias [56].

CEA-Leti of France has been making HgCdTe APDs for space lidars and optical communications [57,58]. They have a spectral response from 0.8 to 3.3 μm. The active area is relatively large, 150–200 μm in diameter. The devices can be operated at 160 K with a four-stage TEC, though the device performance is much better at cryogenic temperatures. The electrical bandwidth of the current devices is 180 MHz. The APD gain is 300 at 12 V bias. The NEP at the maximum APD gain is 10–15 fW/Hz^1/2^. An earlier device at 185 K has been evaluated for 2 μm wavelength CO_2_ lidar application and showed clear advantages over InGaAs APDs at this wavelength [59].

Raytheon Vision System also developed high-speed HgCdTe APDs for lidar applications [60,61]. These are planner devices hybridized with a ROIC. The pixel size is about 50 μm with a 100% fill factor. Different sizes of pixel arrays have been made. The cutoff wavelength is designed to give optimal performance at a 1.55 μm wavelength. The maximum APD gain was measured to be about 100, and the electrical bandwidth was greater than 1 GHz from their 4 × 4 pixel arrays.

### 3.4. Radiation Damage and Annealing of HgCdTe APD Arrays

The HDVIP APD arrays were tested with gamma rays up to 100 krad(Si). The APD arrays without the ROIC were tested with protons to 7.5 × 10^11^ protons/cm^2^ fluence (100 krad(Si) ionization dose) [62]. The APD arrays with the ROIC were tested with protons to 30 krad(Si). These proton fluence and ionization doses are many times the likely space radiation exposure for lidar detectors. There were no measurable changes in the APD performance from the gamma-ray irradiation. The major effect of proton irradiation was the increase in the dark current at a rate of about 3 × 10^−22^ A/proton/cm^2^, which started to rise above the noise floor at about 10^10^ protons/cm^2^ (1–2 krad(Si)). Transient protons produced large output pulses equivalent to tens of thousands of photons, which saturated the electronic circuit, for a few μs. It was also found that the dark current increase due to radiation increased by 2–3 orders of magnitude after the APDs were warmed up to room temperature and cooled down again. However, heating the APD arrays at 80 °C for several hours completely annealed the radiation damage and brought the dark current back to the original values. The HgCdTe APD arrays from Leonardo UK were also tested with gamma rays and protons [55]. There were no measurable changes in the APD performance after 29 krad(Si) of gamma-ray irradiation. There were no changes in APD responsivity up to 5.9 × 10^10^/cm^2^ proton fluence. The dark current of the Leonardo UK devices due to proton irradiation was measured by NASA GSFC with the APD array wire-bonded to a ROIC with CTIAs [63]. The dark current from radiation damage also went up significantly after warming up the device to room temperature and cooling it back down again. The dark current returned to the previous level after heating the APD to 70 °C for several hours. Based on these test results, the HgCdTe APD arrays can be used in a typical space radiation environment for a multi-year mission. The APD arrays should be cooled continuously and not be warmed up unless necessary. There should be provisions to heat the APD array to 70–80 °C for a few hours when the dark current becomes high.

### 3.5. Use of HgCdTe APD Arrays in Airborne and Future Spaceborne Lidars

HgCdTe APD arrays were first used in an airborne continuous wave (CW) IPDA lidar for carbon dioxide (CO_2_) measurements at 1.57 μm laser wavelength in 2013 [64]. The device was an 8 × 8 pixel array from Leonardo DRS. The outputs from individual pixels were connected in parallel and then to a low-noise preamplifier outside the Dewar. Although the overall receiver noise was high due to the large stray capacitance, the HgCdTe APD array was instrumental in the successful demonstration of CO_2_ measurement at 1.57 μm from an airborne platform.

The 4 × 4 pixel HgCdTe APD arrays were successfully used in the airborne pulsed CO_2_ IPDA lidar at 1.57 μm laser wavelengths in 2014, 2016, and 2017 [65], and an airborne methane (CH_4_) IPDA lidar at 1.65 μm laser wavelengths in 2017 [66]. The APD arrays were housed in closed-cycle cryo-coolers and cooled to 80 K. These detectors greatly improved the receiver performance compared to those with the near-infrared PMTs used earlier. The receiver linear dynamic range spanned four orders of magnitude, which was sufficient to measure aerosol backscatter and much stronger ground returns at a fixed APD gain and TIA gain setting. A similar detector system has been used for an airborne IPDA CO_2_ lidar at 2.05 μm laser wavelength and showed a major improvement in the receiver performance over the InGaAs APDs [67].

The 4 × 4 pixel HgCdTe APD arrays have also been used for prototype planetary science lidars. One such detector system was used in the development of a multifunction climate lidar for Mars to measure wind and atmospheric backscatter profiles from orbit [68]. A 4 × 4 pixel IDCA is being used in the Spectroscopic Infra-Red Reflectance Lidar (SpIRRL) to measure water ice on the Moon in the 3 μm water band. A 2 × 8 pixel LMPC IDCA has been used in a prototype lidar instrument for small airless bodies such as asteroids and comet cores [69].

## 4. Discussion

Photodetectors play a critical role in space lidars where the signal is weak, and the operating environment is severe. It is difficult and expensive to develop space-qualified detectors that meet both the performance and reliability requirements. There have been only a few types of photodetectors used in the past and present space lidars. Ruggedized PMTs have been used in UV and visible wavelengths. Actively quenched Geiger mode silicon APD SPCM from Excelitas Canada have been used in atmospheric lidars at visible wavelengths. The SPADs from the Czech Technical University have been used in a planetary lidar and space laser ranging instrument. The infrared QE-enhanced silicon APDs in analog mode from Excelitas Canada have been used for 1.064 μm wavelength in almost all space lidars in the past 30 years.

HgCdTe APD arrays have emerged recently to fill the gap in single photon detectors from near- to mid-infrared. The HgCdTe APD arrays from Leonardo DRS are the most mature at present in terms of performance and packaging, but devices from other manufacturers are rapidly improving. The technologies for small cryo-coolers in space are also mature. IDCAs with HgCdTe APD arrays have been successfully built and tested. The 4 × 4 pixel HgCdTe APD arrays developed by Leonardo DRS for NASA have been successfully used in several airborne atmospheric lidars. A 2 × 8 pixel LMPC HgCdTe APD array has been integrated into prototype surface elevation lidars for small airless planetary bodies. Larger size arrays with more pixels have been fabricated and tested in the laboratory. Space radiation effects on HgCdTe APDs have been investigated, and an effective annealing technique for radiation damage has been found.

HgCdTe APD arrays have extended lidar measurement capabilities from near- to mid-infrared for active spectroscopy measurements. They bring the receive sensitivity to near the quantum limit to allow the use of low pulse energy lasers for long-distance measurements. They can be made into large, multipixel arrays to map surface topography with wider spatial coverage at finer resolution. They give linear waveform outputs at a sub-photon noise level to measure the vertical profile of vegetation and atmospheric backscattering. HgCdTe APD arrays have been successfully integrated with tactical coolers and integrated into prototype lidar instruments that are close to being space-qualified. The near-quantum limited performance and extended spectral response from the HgCTe APDs will likely have a major impact on space lidar deployment for future Earth science and planetary science missions.

## Figures and Tables

**Figure 1 sensors-24-06620-f001:**
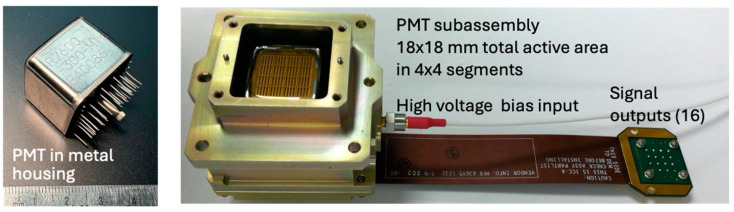
Photographs of the PMT used in ICESat-2/ATLAS. They were made by Hamamatsu Corporation, Model R7600-300-M16, and up-screened and space-qualified at NASA. The PMT consists of an extended bi-alkali photocathode, a 10-stage dynode chain, and 4 × 4 segmented anodes. There are 16 outputs, one for each anode.

**Figure 2 sensors-24-06620-f002:**
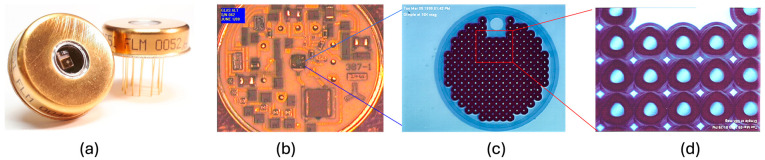
Photographs and micrographs of the silicon APD preamplifier module used in GLAS on ICESat: (**a**) detector module in a 1-inch diameter TO-8 invar metal housing with an optical window; (**b**) hybrid circuit containing an APD chip, a preamplifier and a high voltage bias regulation circuit; (**c**) top surface of the active area of the APD with an array of small dimples to deflect the light to increase the photon absorption path length; and (**d**) enlarged view of the dimple array.

**Figure 3 sensors-24-06620-f003:**
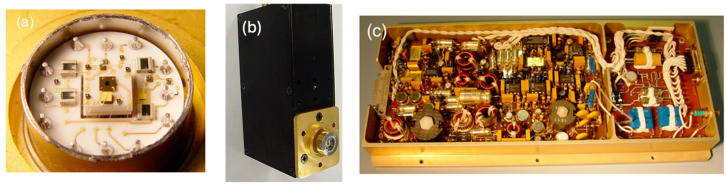
Photographs of an SPCM used in ICESat/GLAS: (**a**) detector header with the window removed, showing the silicon APD on top of a TEC and the first stage quenching electronics on a hybrid circuit; (**b**) SPCM subassembly used in ICESat/GLAS; (**c**) the power supply and TEC control electronics made by Space Power Electronics, Inc., Bonaire, GA, USA.

**Figure 4 sensors-24-06620-f004:**
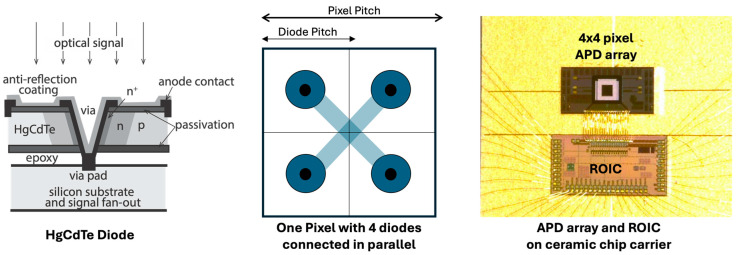
Illustration of HDVIP HgCdTe APD layout and a photograph of the 4 × 4 pixel HgCdTe APD array and ROIC on the chip carrier.

**Figure 5 sensors-24-06620-f005:**
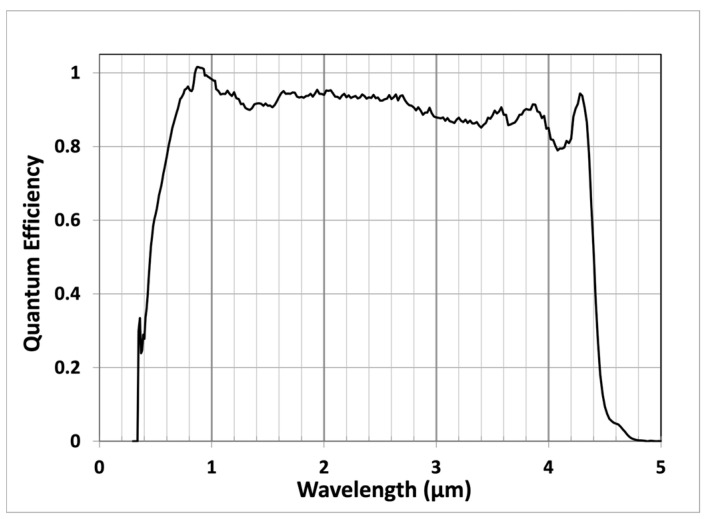
Spectral response of the 4 × 4 pixel HgCdTe APDs [46]. The QE is defined as the ratio of the number of output photoelectrons to the number of incident photons at unity APD gain.

**Figure 6 sensors-24-06620-f006:**
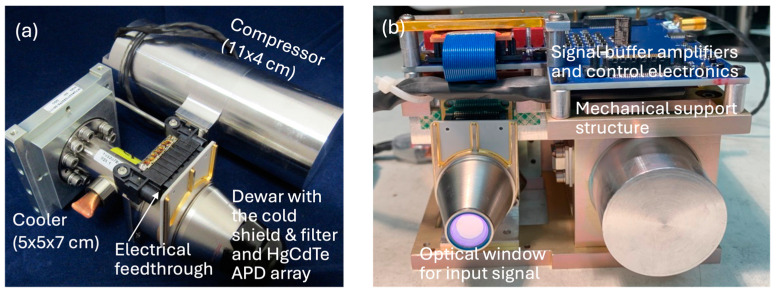
Photographs of the IDCA with the 4 × 4 pixel HgCdTe APD array in a tactical cooler: (**a**) tactical cooler with the APD array in the Dewar before integration with the external electronics; (**b**) complete IDCA with the signal buffer amplifiers and detector control electronics on an aluminum support structure.

**Figure 7 sensors-24-06620-f007:**
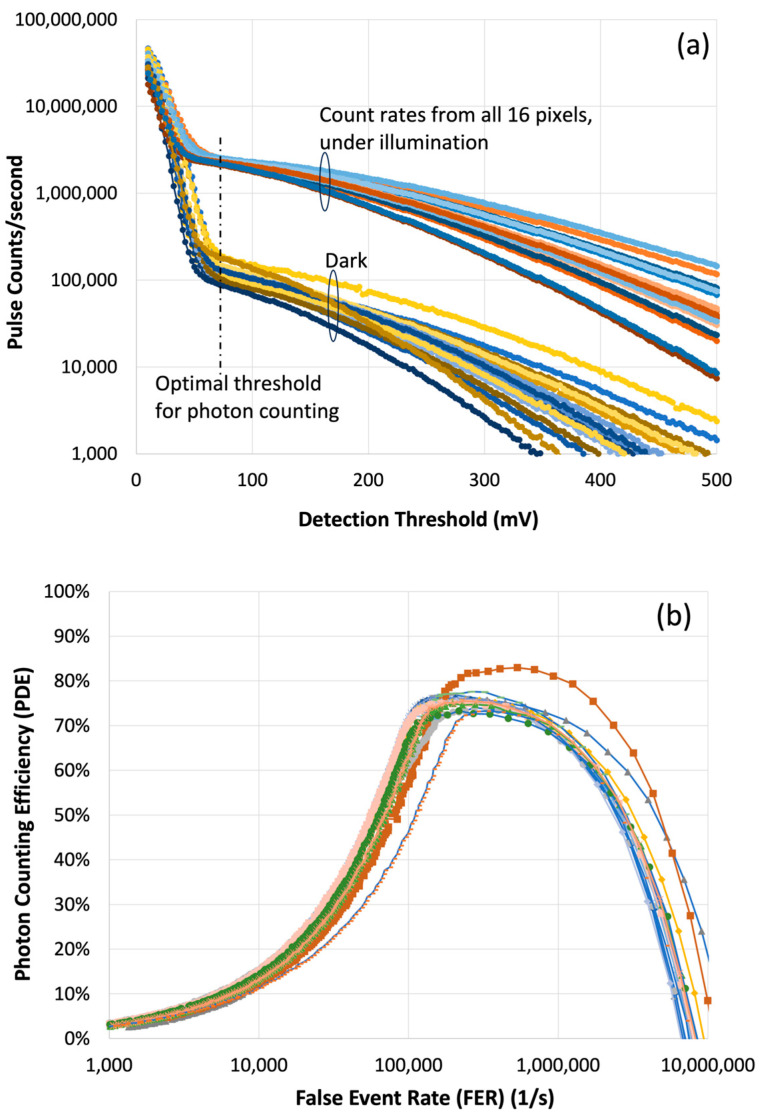
Test results of the 2 × 8 pixel IDCA: (**a**) pulse counts vs. detection threshold from individual pixels in the dark and under illumination at 1.03 μm wavelength; and (**b**) PDE vs. FER at different detection thresholds. The colors of the curves indicated the outputs from different pixels. The PDE is the ratio of the rate of the net detected photons to the rate of the incident photons. The FER is the rate of the dark counts.

**Figure 8 sensors-24-06620-f008:**
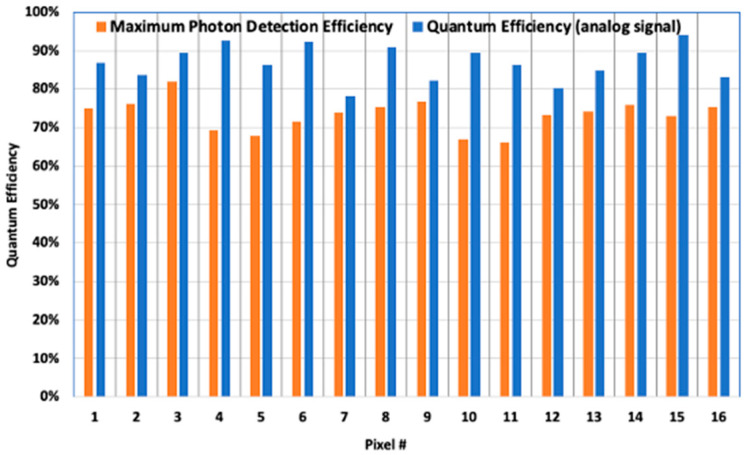
QE of the analog signal output and maximum PDE from threshold crossing detection of the latest 2 × 8 pixel IDCA measured at 1.55 μm wavelength.

**Figure 9 sensors-24-06620-f009:**
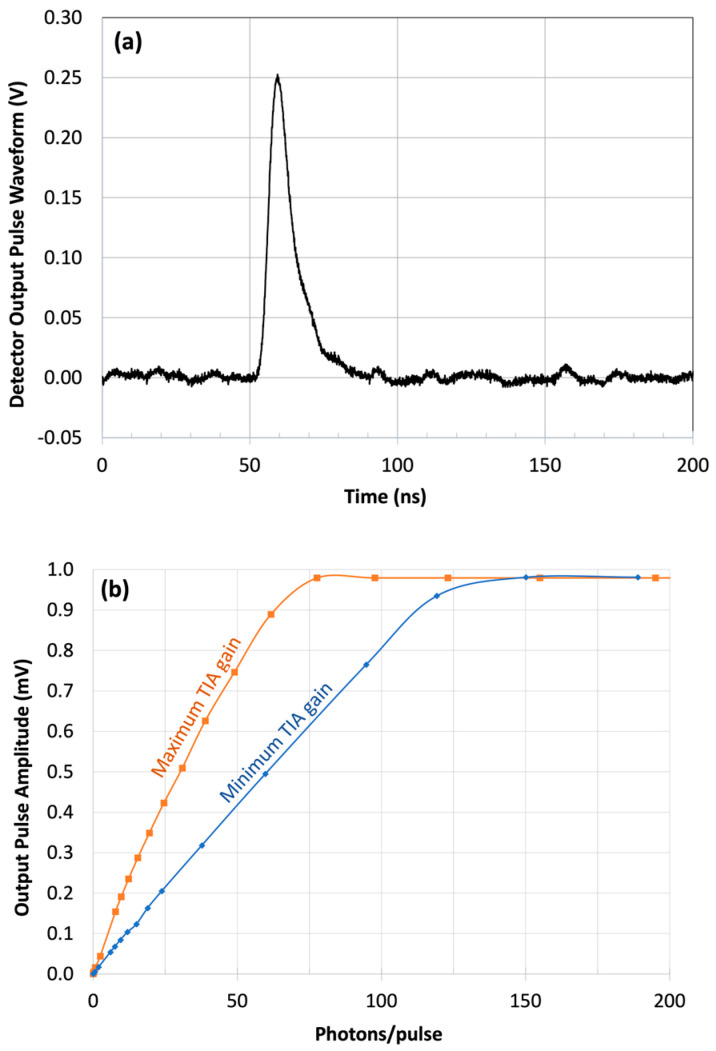
(**a**) Impulse response pulse shape; (**b**) output pulse amplitude vs. the input pulse energy in photons/pulse from the 2 × 8 pixel IDCA at 1.55 μm wavelength.

**Figure 10 sensors-24-06620-f010:**
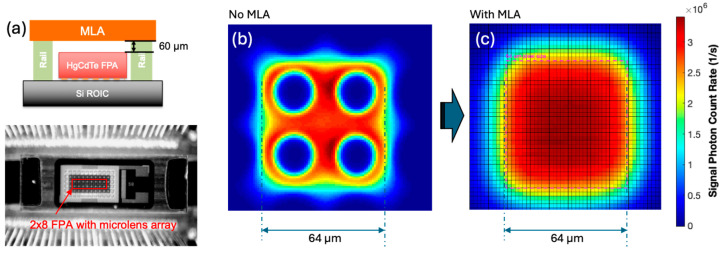
(**a**) Illustration and photograph of the microlens array (MLA) above a 2 × 8 pixel HgCdTe APD array which is also known as focal plane array (FPA); (**b**) surface scan of a single pixel without the microlens array; and (**c**) surface scan with the microlens array. The surface scan was performed at 1.03 μm wavelength with a light spot size of about 5 μm.

**Figure 11 sensors-24-06620-f011:**
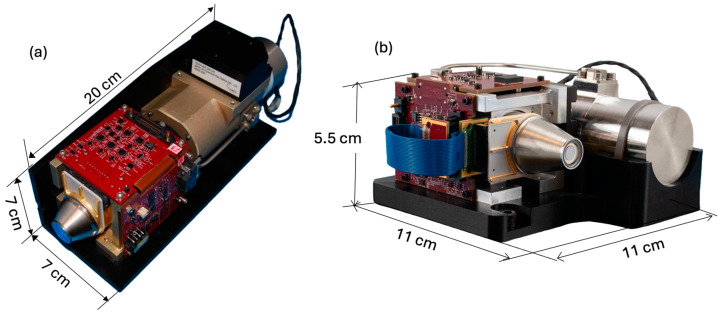
Photographs of the 2 × 8 pixel IDCAs: (**a**) the first generation IDCA designed for CubeSat [51]; (**b**) the latest IDCA designed for use in a small lidar. The IDCAs weigh about 0.8 kg with all the peripheral electronics. The cryo-cooler itself consumes about 6 W of electrical power. The electronics consume less than 2 W of power.

**Table 1 sensors-24-06620-t001:** Characteristics of the 4 × 4 pixel HgCdTe APD arrays.

Parameters	Measurement Results
Quantum Efficiency	>90%, 0.9 to 4.4 μm
APD gain	1 to 900
Bias voltage	0 to 12 V
Excess noise factor	1.05
Dark current	<0.5 pA/pixel
Maximum TIA gain	320 kV/A
Responsivity	>2 × 10^9^ V/W
Electrical bandwidth	8 MHz
NEP	<0.5 fW/Hz^1/2^/pixel at 1.55 μm wavelength and 12 V APD bias
Maximum no damage input optical power	tested to >37 μW/pixel, at 1.55 μm
Pixel size	80 × 80 μm
Pixel pitch	80 μm
Fill factor	75%
Operating temperature	77–120 K

**Table 2 sensors-24-06620-t002:** Characteristics of the 2 × 8 pixel IDCA.

Parameters	Measurement Results
Quantum Efficiency	>90%, 0.9 to 4.4 μm
APD gain	1 to 1900, with APD bias from 0 to 12 V
Excess noise factor	1.15
Outputs	16 individual outputs for 2 × 8 pixels
Dark current	<8 fA (50,000 electrons/s) per pixel
TIA gain	150 to 250 kV/A
Buffer amplifier gain	8 V/V
Electrical bandwidth	50 MHz
Impulse response	6–9 ns FWHM
Single photon pulse time jitter	<1 ns
Responsivity at 1.55 μm	1.0 to 1.5 × 10^9^ V/W with buffer amplifiers
Single photon pulse amplitude	>25 mV
NEP at 1.55 μm	<0.2 fW/Hz^1/2^/pixel
Pixel size	64 × 64 μm
Pixel pitch	64 μm
Fill factor	100% with microlens array
IDCA size	11 × 11 × 5.4 cm
Cold shield aperture	f/7 or f/3
Cryo-cooler power	28 V, 6 W with heat sink at room temperature
Operating temperature	80–110 K
Electrical power	<2 W, +5 V, +3.6 V, and −2 V.
Operation temperature	−24 to 60 °C
Storage temperature	−40 to 71 °C

## Data Availability

Data are contained within the article.

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
