# Peer review of "Review of Photodetectors for Space Lidars"

_sensors, 2024, doi:10.3390/s24206620_

Round 1
Reviewer 1 Report
Comments and Suggestions for Authors
This manuscript provides a comprehensive review of photodetectors used in space lidars, focusing on their sensitivity, spectral response, and radiation tolerance. The paper thoroughly discusses the evolution of photodetector technology from early single-element devices to modern pixelated arrays, highlighting the advancements in HgCdTe avalanche photodiode arrays for future space lidar applications. However, the manuscript would benefit from significant revisions to enhance its comprehensiveness and clarity.
1. The review primarily concentrates on infrared and visible light detectors, yet there is a gap in discussing the progress of ultraviolet detectors in the field of spaceborne remote sensing. It is recommended that the authors include a section on the current state and challenges of ultraviolet detectors, addressing the specific bottlenecks they face in long-distance space applications.
2. While the paper compares various detector technologies, a more detailed comparative analysis is needed, particularly focusing on performance metrics such as quantum efficiency, dark current, and dynamic range. Including a table or chart that juxtaposes these parameters for different detector types would greatly benefit the reader.
3. The sentence "The performance of linear mode APDs is still limited by the electronics noise of the subsequent circuit, the dark noise of the photodetector, and the randomness of the APD gain." is somewhat ambiguous and could be rephrased for clarity. The authors should also undertake a thorough language review to ensure all sentences are grammatically correct and unambiguous.
4. For references, it is recommended to add the following references to expand the introduction for attracting the electronic community's interest:
Zhu, S.; Lin, Z.; Wang, Z.; Jia, L.; Zhang, N.; Zheng, W. PhotoniX 2024, 5, (1), 5.
Comments on the Quality of English LanguageNA
Author Response
See the attached file "ResponseToReviewer-1.docx"

Reviewer 2 Report
Comments and Suggestions for Authors
Although the reviewer is not a familiar with space lidars, they are very familiar with photodetectors. The review sets out a useful background and introduces the HgCdTe APD and its future applications. It is a valuable contribution.
The reviewer read the paper with great interest. However, there are significant English and accuracy errors, which suggests a lack of attention to detail which questions whether this is careful scientific work. Somes figures in particular are of poor quality and need to be improved.
I recommend publication after the following issues are addressed
Science:
They have high and almost noiseless APD gain to completely override the electronics noise without excess noise. Line 59. This is unclear.
They have demonstrated near quantum limited receiver performance. Line 60. Needs a reference.
Line 70 needs referenced.
Line 85 needs referenced – long-time isn’t really specific enough.
QEs need referenced in lines 87-89.
Why can PMTs in analogue mode not have deadtime? Surely there is still a recovery time or am I mistaken? – line 110.
Line 195 – are they not heated to 70 K not C?
What is condensing laser light (why is it different from focusing) line 308.
Figure 5 is exceptionally poor with random ? dotted around. Seems lazy.
Figure 6 is quite unclear and unhelpful in demonstrating the above points.
Figure 7 is valuable but presented very badly.
Figure 9A is incredibly unclear. I can hardly see the purple line. This is probably unacceptable for this journal.
The author has described in some detail the photon capture but neglected to discuss the state of the art in timing circuits – all of which the described technologies are reliant on. Might there be scope to include some discussion on the actual counting electronics?
Comments on the Quality of English LanguageThe English needs to be improved. I have supplied a non-exhaustive list of suggested corrections:
In abstract:
avalanche photodiode arrays have emerged recently to fill *these* needs.
*A* linear response is needed to give range resolved atmospheric backscatter and surface reflectance.
Wide dynamic range is required to accommodate the received signal from atmospher*ic* backscatter
For topographic measurement, multi-pixel photodiode arrays are needed to map the surface without gaps along and cross ground track. *along with what?*
Perido (spelling mistake – line 52)
SWIR wavelength (should be wavelengths – line 63).
Line 207 – device not devices
Line 222- degreation not degredations
Line 274 – MPD name is wrong.
Line 335 – detect single photons
Line 577 damage not damges.
Author Response
See the attached file "ResponseToReviewer-2.docx"

Round 2
Reviewer 2 Report
Comments and Suggestions for Authors
I am thankful for the author to spend the time to improve the manuscript!
I am happy to recommend acceptance if the following is addressed:
Figure 5 - The spectral response cannot go above 1 in quantum efficiency without explanation. Maybe the author wishes to add an error? I cannot recommend acceptance until we make sure everything is physical!
English is much improved. Some minor corrections:
line642
space radiation effects
Line 662
thank Jim Abshire of NASA GSFC for his constant encourageMENT for me to work on space lidar
Comments on the Quality of English LanguageSee above
Author Response
Thank you again for reviewing the revised manuscript. Your comments and suggestions on both rounds of peer reviews have been extremely helpful for me to improve the manuscript. I also learned the lesson to always be diligent when it comes to scientific writing and spend extra time to improve my English. Please find the my detailed response in the attached file "ResponseToReviewer-Round2.docx".
Sincerely,
Xiaoli
